# Sense-checking the approach to quantitative sensory testing to detect chemotherapy-induced peripheral neuropathy

Marin Dujmović[1]*, James Philip Dunham[1], Johannes Gausden[2], Bethany Groves[1], Elizabeth Anne Cottuli de Cothi[1], Charlotte Burgess[1], Chris Geddie[1], Lottie Adams[1], Alex Grant[1], Nisla Maharjan[1], Bhushan Thakkar[3], Anthony O'Neill[2], Alan Young[4], Roger Graham Whittaker[5], Lesley Colvin[3], Anthony Edward Pickering[1]

1 Anaesthesia, Pain and Critical Care Research, School of Physiology, Pharmacology and Neuroscience, University of Bristol, Bristol, United Kingdom, 2 School of Engineering, Newcastle University, Newcastle upon Tyne, United Kingdom, 3 Division of Population Health and Genomics, School of Medicine, University of Dundee, Dundee, United Kingdom, 4 Patient Partner, SenseCheQ Project, Bristol, United Kingdom, 5 Translational and Clinical Research Institute, Newcastle University, Newcastle upon Tyne, United Kingdom

* marin.dujmovic@bristol.ac.uk

## Abstract

People with chemotherapy-induced peripheral neuropathy (CIPN) have abnormalities in Quantitative Sensory Test (QST) findings. However, the predictive utility of QST for the early detection of CIPN in individuals has not been demonstrated. This will require longitudinal QST during chemotherapy treatments. However, QST is time-consuming, requires expertise and complex, costly equipment which has largely prevented its adoption in routine clinical practice. We aimed to assess approaches to develop a reliable, straightforward, time efficient and sensitive sensory testing method. Guided by patient partner input and previous literature, we selected thermal and vibration detection thresholds as target QST parameters. A series of iterative experiments was conducted to determine the optimal body test site and to develop a novel vibration testing protocol. The thenar eminence emerged as the best candidate due to higher sensitivity to all stimulus modalities, lower variance and less age-related change compared to the feet. We demonstrated significant differences in thermal thresholds between healthy participants and people with CIPN measured at the thenar but not the feet. The vibration testing protocol, employing a linear resonant actuator, performed better than a calibrated tuning fork being sufficiently sensitive to identify age-related and body site differences in sensory function well as tracking sensory loss induced by local anaesthetic nerve block. These findings establish a testing framework to deliver QST at a single convenient body site with a reduced set of modalities to efficiently track multifibre sensory function for patients at risk of developing neuropathy.

**Data availability statement:** Anonymized data and R markdown code for reproducing analyses is available on the Open Science Framework at http://doi.org/10.17605/OSF.IO/F5T82.

**Funding:** The SenseCheQ research project was funded as part of the Advanced Pain Discovery Platform by a consortium made up of UKRI MRC, Arthritis UK, and Eli Lilly all coordinated by the MRC [MR/W027925/1]. BT was also supported by an APDP grant as part of the Partnership for Assessment and Investigation of Neuropathic Pain: Studies Tracking Outcomes, Risks and Mechanisms (PAINSTORM) consortium [MR/W002388/1].

**Competing interests:** AEP declares consultancy work for Lateral Pharma and has a research grant from Eli Lilly on an unrelated topic. All other authors have no conflicts of interest to declare.

# 1. Introduction

Chemotherapy-induced peripheral neuropathy (CIPN) develops as a side-effect of many effective and commonly used systemic anti-cancer agents (notably platins [1,2], taxanes [3,4], bortezomib [5,6], and vinca-alkaloids [7,8]). The resulting nerve damage can severely impact quality of life causing neuropathic pain, impaired dexterity or balance and autonomic problems [9,10]. About 68% of people will experience acute CIPN during chemotherapy treatment and it will become chronic for 30% (lasting more than six months after treatment ends) [10].

There is no effective treatment to attenuate the symptoms or reverse this nerve damage, but detection in the acute stage of neuropathy can lead to changes in chemotherapy regimen to reduce the probability of developing chronic CIPN [9]. However, the early detection of CIPN is difficult. In most cases it depends upon early patient report, which is problematic because symptoms tend to be insidious and ambiguous leading to delays in recognition, when it can be too late to intervene [11]. Patients report a lack of familiarity with CIPN, and they can be reticent about reporting symptoms for fear of changes to their treatment and perceived reduction in the likelihood of cure [12,13]. Formalised assessments (e.g., Total Neuropathy Score [14,15], National Cancer Institute Criteria for Adverse Events [16,17]) and patient report questionnaires (e.g., EORTC-CIPN-20 [18,19]) are used in research to determine whether patients have developed CIPN. However, they are variably deployed in clinical practice, their validity as reliable tools has been repeatedly questioned, and there is currently no gold standard diagnostic criterion for CIPN [20].

Attempts to develop more objective measures have included the use of quantitative sensory testing (QST) using standardized protocols [21–25]. Multiple studies have identified differences in sensory function after systemic anti-cancer treatment or in patients diagnosed with CIPN [26–34]. This suggests QST has potential utility in diagnosis of CIPN and there is evidence that tests of thermal and mechanical detection thresholds are the most sensitive indicators of neurotoxicity [34]. However, there is little evidence to indicate that QST can be deployed for early detection of the neurotoxicity leading to CIPN. In part, this is a consequence of the challenge of QST in longitudinal studies as it is demanding for patients, time consuming, requires expertise to deliver and interpret and typically utilises complex, clinic-based equipment. The same considerations have prevented its incorporation into routine clinical practice during treatment.

The aim of the SenseCheQ project is to establish a reliable, easy to use approach for self-administered QST. This could allow patients to monitor their nerve health regularly during treatment in their homes. Attempts have been made to develop bedside QST [35–39] but they show low sensitivity and are not designed for self-administration by patients so are unsuitable candidate approaches for reliable testing in the community. Developing such an approach poses a number of technological, methodological and pragmatic questions to address the brief set by our patient partners. We aimed to assess approaches to develop a sensitive, reliable, straightforward, and time efficient sensory testing approach including comparisons between healthy controls and people with CIPN.

## 2. Methods

### 2.1. Patient partner input

Our SenseCheQ patient partner group (N=6), who all had lived experience of CIPN, guided us with their insights and feedback throughout the project from inception to publication. They co-developed our patient facing information, experimental design and recruitment strategies, helped with the interpretation, preparation and dissemination of our findings including co-authorship of this publication (AY). Their insights motivated our experiments as indicated throughout the results text (*in italics*). We met with the PPI group throughout the project (every 3–4 months) and the PPI lead attended all our monthly team meetings and facilitated the information transfer regarding project updates and feedback on research findings.

*In initial PPI discussions about developing an alternative sensory testing approach they emphasised the importance of accessibility and simplicity. This included conducting sensory testing at a single body site. The group strongly recommended that testing for more than 10–15 minutes per session would be likely to deter patients from doing so regularly, in the context of their ongoing treatment. This resulted in an evaluation of a reduced set of QST parameters.*

### 2.2. Ethics, participants and study environment

Experiments were conducted with adults (≥18 years old): either healthy volunteers or people with CIPN. For all healthy participants (recruited among students and staff at the University of Bristol) the same exclusion criteria applied: acute or chronic pain conditions; regular or recent (within 24 hours) analgesic use; presence of neuropathy; broken or infected skin at the testing areas, or allergies to latex and alcohol. Patients who had completed chemotherapy treatment for cancer at the Bristol Haematology and Oncology Centre (University Hospitals Bristol and Weston) and were identified as having CIPN (diagnosed by their clinical team) were referred to the study by their clinical care team or by contacting us directly (poster and leaflet advertisements at the oncology centre). After initial contact patients were given further information (via participant information sheet) and the same exclusion criteria were applied (with the exception of neuropathy/analgesic use).

Ethical approval was obtained from the University of Bristol Research Ethics Committee for healthy participant studies (project ID:9994). For patient studies, ethical approval was obtained from Health Research Authority and Health and Care Research Wales (IRAS project ID: 327541). All study participants provided written informed consent. Experimental sessions were conducted in a controlled clinical study room environment (at the NIHR Bristol Clinical Research Facility) with ambient temperature of 21°C. In all studies participants were reimbursed for their time at a rate of £10/h.

### 2.3. Body site and sensory modality selection

Due to the glove and stocking pattern of symptoms of CIPN, the options for sensory testing to detect the development of neuropathy were restricted to the distal limbs (feet and hands). *Our patient partner group suggested that we should avoid pain threshold testing if at all possible for the home sensory testing protocol.* We agreed as firstly, pain threshold tests have been shown to have high intra-individual variability [25]. Secondly, these tests take longer within standard QST protocols (particularly cold pain testing at 1°C/s) and we aimed to keep the entire protocol as short as possible to encourage regular self-administered testing (less than 15 minutes per patient partner input). Finally, and most importantly, they are less sensitive at identifying CIPN than detection thresholds [34]. Taking this into account, we chose to focus on warm (WDT), cold (CDT), and vibration detection (VDT) thresholds from the standard protocol developed by the German Research Network on Neuropathic Pain (DFNS) [25]. Measuring these thresholds allowed the function of C-, Aδ- and Aβ-fibres to be tested, respectively [34].

#### 2.3.1. Sensory testing on the sole vs dorsum of the foot.
Warm and cold detection thresholds (WDT & CDT, as well as heat pain) were measured using a 9 cm² thermal probe (Medoc TSA-II) using the standard DFNS protocol

[25] (stabilizing skin temperature to 32˜C as baseline and ramping at 1°C/s). Vibration detection thresholds (VDT) were measured with a 64 Hz Rydel-Seiffer calibrated tuning fork. One familiarization trial was followed by 3 test trials. Mechanical detection and pain thresholds were measured using Semmes-Weinstein monofilaments (Touch-Test®) and calibrated pinpricks (MRC Systems GmbH) respectively. Mechanical detection was conducted by determining the first filament weight which did not elicit a touch percept, then proceeding in a staircase manner. The procedure ended when five sub- and supra-thresholds values were measured. Mechanical pain was measured in a similar manner; the first pinprick weight that elicited a sharp percept was determined followed by a staircase approach to establish five sub- and supra-threshold values. All participants underwent the same order of testing of their non-dominant side, first at the dorsum (participant supine) and then at the ball of the foot (participant prone).

**2.3.2. Thermal sensitivity of dorsum of the foot vs thenar eminence.** Thermal detection thresholds (CDT and WDT) were measured (QST.Lab TSC II with the T09 thermal probe). Tests were performed with 2 of the 5 Peltier elements on the probe (12x15mm) to ensure that the probe elements were reliably centred in contact with the thenar eminence even in participants with smaller hands. The protocol was the same as for CDT and WDT in **2.3.1**. Participants were first tested at the thenar and then the foot on their non-dominant side.

**2.3.3. Thermal sensitivity in people with CIPN vs healthy controls.** CDT and WDT were measured at the thenar eminence and dorsum of the foot of the non-dominant side for people with CIPN and matched healthy controls (from **2.3.2**, using the same test protocol).

## 2.4. Haptic actuator testing and validation

The Rydel-Seiffer calibrated tuning fork has been used as the standard for vibration detection testing and is part of the DFNS standardized protocol. However, the calibrated tuning fork is rarely used in clinical practice and is not suitable for self-test assessments by patients. We aimed to develop a haptic stimulator which provides a controlled oscillation whose amplitude can be continuously varied, that performs well at low amplitudes and has a small footprint. We employed a linear resonant actuator (HapCoil-One, Actronika, Paris, France; 11.4x12x37.7 mm, 8.7g) which operates within a wide range of frequencies (10–1000 Hz; resonant frequency of 65 Hz).

**2.4.1. Age and body site differences in vibration sensitivity.** The haptic actuator was enclosed within a 3D printed box enclosure (55x30x20 mm) and controlled using an Arduino MKR Zero micro-controller (see Fig 1A). This was used to deliver precise sine wave oscillations with ramps of ascending or descending amplitudes at a range of frequencies. In this study we chose 64 Hz in order to match the calibrated tuning fork. The axis of oscillation was oriented in a plane parallel to the surface of the skin. Ascending and descending ramping stimuli were delivered in the range from 0 (no vibration) to 0.30 (maximum) – these are arbitrary units (AU) which are linearly related to vibration amplitudes as measured by an attached accelerometer. In the configuration used for this experiment the maximum amplitude displacement was approximately 20 μm, which is about five times higher than the expected thresholds for healthy adults at the dorsum of the foot (the less sensitive of the two locations) [40]. In ascending ramps participants indicate when they detect vibration and in descending ramps they indicate when they feel the vibration stop. The calibrated tuning fork delivers a descending vibration stimulus – at a high amplitude once struck and the amplitude decays exponentially. With the tuning fork, participants report when they feel the fork stop vibrating and the experimenter reads the value on a 1–8 scale (higher values indicate lower vibration amplitudes, i.e., higher sensitivity to vibration). Vibration detection thresholds were measured using ascending and descending ramps with the haptic testing apparatus and then with a tuning fork at the dorsum of the foot and then thenar eminence. For each method, one familiarization trial was followed by 3 measured trials.

**2.4.2. Detection of sensory loss with a haptic actuator and tuning fork.** As a model of sensory loss, lidocaine was injected subcutaneously (3 ml of 1%) between the medial and lateral malleoli to produce a field block of the superficial peroneal nerve (SPN) over the forefoot (Fig 1B and 1C). Sensory tests were conducted before block and repeated at 20,

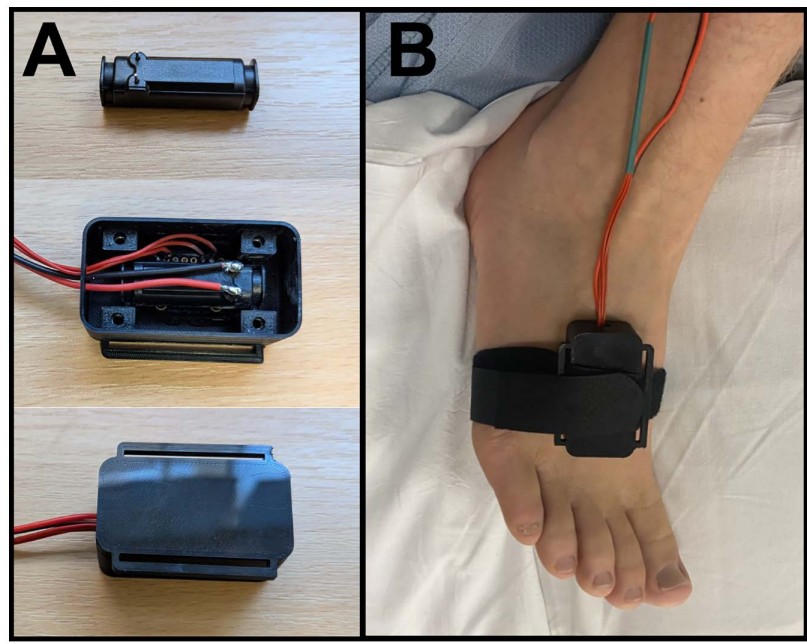

**Fig 1. Encased Hapcoil One and site of vibration detection measurement.** A – Hapcoil One haptic actuator and 3D printed enclosure; B – Enclosure at the dorsum of the foot.

40 and 60 minutes after the lidocaine injection. The development of the analgesic block was assessed using a 512 mN pinprick stimulator (MRC systems, Germany, one familiarization stimulation followed by 3 measured stimulations) which participants rated on a 0–100 pain rating scale (0 – no pain; 100 – worst pain imaginable). VDTs were measured using the haptic actuator and the calibrated tuning fork (as per **2.4.1**). The tests were repeated in a set sequence: ascending and descending thresholds using the haptic actuator (see Fig 1) followed by the tuning fork and then pain ratings for pinprick stimuli.

## 2.5. Data analysis

Thermal and vibration thresholds for each participant in all experiments were computed as means across trials. Mechanical detection and pain thresholds were computed as geometric means. Parametric tests such as t-tests and ANOVAs were planned alongside appropriate assumption tests (assessment of normality of residuals, equality of variance, sphericity as required). Where assumptions were violated then data transformations were considered (e.g., log-transform). Where data transformation failed to adequately account for failed assumptions non-parametric tests were selected as appropriate. Bonferroni corrections were applied for post-hoc tests to control for family-wise error rates. For pair-wise comparisons of all kinds, we report either mean difference between groups/conditions ($M_{diff}$) with a 95% CI for the estimated difference, or an estimated difference between medians ($ME_{diff}$) with the associated 95% CI (the Hodges-Lehmann estimate).

## 2.6. Power analysis

This project involved the iterative development and testing of novel approaches to QST for which the sensitivity and hence effect size was unknown in many cases. This precluded a formal power calculation for several of the experiments; instead we present results of post-hoc sensitivity analyses which reveal the smallest effects we were powered to detect (Table 1). All sensitivity power analyses were conducted at an alpha of 0.05 and desired power level of 0.80 (using G*Power

**Table 1. Results of sensitivity power analyses.**

| Analysis | Statistical test | Power to detect |
|---|---|---|
| **3.1.1.** Sole vs Dorsum foot | Paired t-test | $d \geq 0.66$ |
| **3.1.2.** Dorsum vs Thenar | 2-way ANOVA | $\eta_p^2 \geq 0.25$ |
| | + post-hoc tests | $d \geq 1.11$ |
| **3.1.3.** CIPN vs controls | Mann-Whitney U-test | $r \geq 0.51$ |
| **3.2.1.** Actuator vs Tuning fork | 3-way ANOVA | $\eta_p^2 \geq 0.17$ |
| | + post-hoc tests | $d \geq 0.89$ |
| **3.2.2.** Detecting sensory loss | 2-way ANOVA | $\eta_p^2 \geq 0.46$ (Ramp type) |
| | | $\eta_p^2 \geq 0.17$ (other effects) |
| | + post hoc tests | $d \geq 0.89$ |

v3.1.9.4 [41]). Where relevant the analyses were done for two-tailed tests and for ANOVAs non-centrality calculation options were set using the option *"as in SPSS"*.

## 3. Results

### 3.1. Body site selection for sensory testing

*Our PPI group suggested that the sole of the foot could represent a convenient choice of test site compared to the dorsum of the foot as patients could simply rest on a test pad. This also makes it easier to remove the foot from the applied stimulation in case of discomfort. Resting the foot on a test pad also would not require fastening of the thermode compared to the dorsum which they noted to be potentially challenging for patients to do on their own.*

**3.1.1. Sensitivity at foot dorsum vs sole.** We conducted a within participant study comparing the sole and dorsum of the foot to assess their relative sensitivity to thermal and mechanical stimulation. A total of 20 healthy young participants (14 female; $M_{age} = 23.15$, Min-Max = 19–35) were tested. Participants were significantly less sensitive to temperature changes at the sole of the foot (Fig 2A). The change from baseline needed to be larger at the sole for participants to detect cold ($M_{diff} = 2.71°C$ [1.72, 3.70], $t(19) = 5.75$, $p < 0.001$, $d = 1.29$), warm ($M_{diff} = 2.22°C$ [0.82, 3.63], $t(19) = 3.47$, $p = 0.003$, $d = 0.78$) and heat pain ($M_{diff} = 2.44°C$ [1.35, 3.53], $t(19) = 4.69$, $p < 0.001$, $d = 1.05$) (paired *t*-tests). There was also a larger variance for CDT and WDT across participants at the sole versus the dorsum of the foot (Fig 2A; CDT 8.43% vs 1.83% and WDT 9.62% vs 3.22%, coefficient of variance).

The sole of the foot was also significantly less sensitive to mechanical pain (Fig 2B, $ME_{diff} = 292.15$ mN [195.52, 406.58], $W = 2.00$, $Z = 3.74$, $p < 0.001$, $r = 0.98$) but there was no significant difference in mechanical detection threshold ($ME_{diff} = 0.11$ mN [−0.30, 2.10], $W = 71.00$, $Z = 0.97$, $p = 0.344$, $r = 0.25$) (Wilcoxon Signed Rank tests). Vibration detection testing using the 64 Hz Rydel-Seiffer calibrated tuning fork showed that all participants scored 8 at both sole and dorsum (maximum sensitivity as measured by the tuning fork).

**3.1.2. Thermal sensitivity at dorsum of the foot vs thenar eminence.** Based on the findings of the previous experiment, particularly for WDT and CDT, we concluded that the sole of the foot did not represent an appropriate site for monitoring sensory function in people with CIPN as the normal ranges showed so much variance even in younger healthy people. *Following discussion of the results with our PPI group they reiterated their concern that self-testing at the feet could be awkward for some patients if they had to strap a device to the top of their foot. This motivated us to compare the sensitivity of dorsum of the foot with the thenar eminence of the hand as a more accessible site for testing.*

We made a comparison in both younger and older participants to determine the sensitivity of the two areas and identify any age-related changes as cancer is more prevalent in the older population. The study groups consisted of 11 younger (5 female; $M_{age} = 27.54$, Min-Max = 21–40) and 17 older (7 female; $M_{age} = 59.59$, Min-Max = 52–69) healthy participants for a

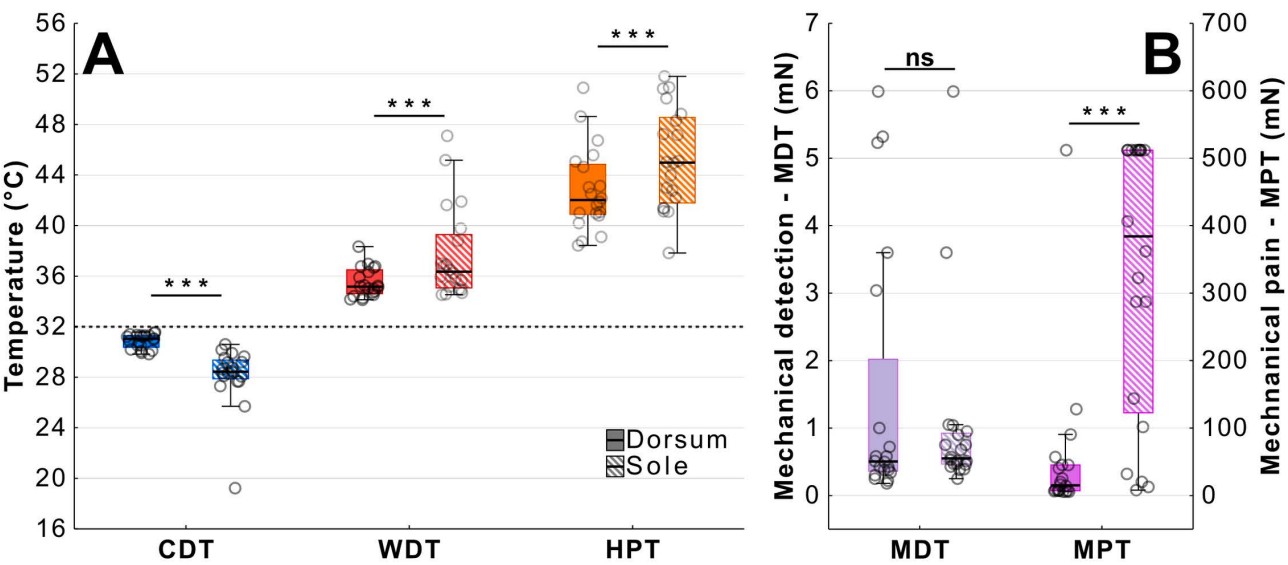

**Fig 2. Comparison of thermal and mechanical thresholds in the foot sole and dorsum.** A – Participants were significantly less sensitive at detecting changes in temperature on the sole for all three measurements. B – The sole was less sensitive to mechanical pain but not to mechanical detection threshold. (CDT – cold detection; WDT – warm detection; HPT – heat pain; Mechanical Detection and Pain – MDT, MPT; **$p < 0.01$; ***$p < 0.001$).

repeated measures study. Two mixed ANOVAs were conducted: one for CDT and one for WDT. One participant from the older group did not feel cold stimulation at the foot at all and was removed from the CDT analysis.

The younger group were more sensitive to cold with higher CDT values ($F(1, 25) = 7.13$, $p = 0.013$, $\eta_p^2 = 0.22$) and CDT values were significantly higher at the thenar eminence rather than the foot ($F(1, 25) = 10.73$, $p = 0.003$, $\eta_p^2 = 0.30$, Fig 3A). Post-hoc comparisons showed that younger participants had higher CDTs at the dorsum of the foot ($M_{diff} = 3.57°C$ [0.28, 6.87], $t(25) = 2.98$, $p = 0.027$, $d = 1.17$) but not at the thenar eminence ($M_{diff} = 0.79°C$ [−2.50, 4.09], $t(25) = 0.66$, $p = 1$, $d = 0.26$) and accordingly the interaction effect was not significant ($F(1, 25) = 2.51$, $p = 0.126$, $\eta_p^2 = 0.09$).

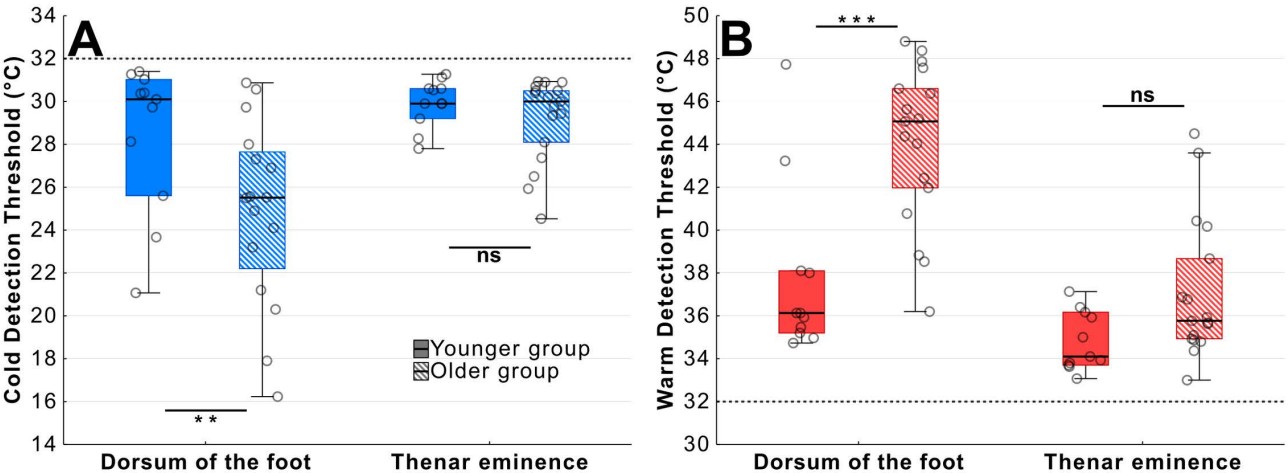

**Fig 3. Influence of test site and age on thermal thresholds.** A – There was a significant difference in cold detection between the two age groups at the dorsum of the foot but not at the thenar eminence. B – Similarly there was a significant difference in warm detection between the two age groups at the dorsum of the foot but not the thenar eminence (**$p < 0.01$; ***$p < 0.001$).

WDT analysis revealed a very similar pattern whereby younger participants were generally more sensitive and had lower thresholds than older participants at the dorsum of the foot ($M_{diff}$=6.25°C [2.70, 9.80], $t(26)$ = 4.84, $p<0.001$, d=1.87) but not at the thenar eminence ($M_{diff}$=2.32°C [−1.24, 5.87], $t(26)$ = 1.79, $p=0.475$, d=0.69; Fig 3B) but this time with a significant age by body site interaction ($F(1, 26)$ = 6.48, $p=0.017$, $\eta_p^2=0.20$).

These results reveal a substantial age-related deterioration in thermal sensitivity at the foot unlike the thenar eminence. The between-participant variance is also much greater at the foot for CDT.

**3.1.3. Thermal sensitivity in people with CIPN vs healthy controls.** We recruited patients clinically diagnosed with CIPN (Table 2) and compared their thermal sensory function with age-matched healthy controls at the thenar eminence and dorsum of the foot to identify between group differences. We were particularly interested to assess whether the known natural deterioration in thermal sensitivity would impact upon the ability to identify significant loss of sensory function in this age demographic.

Patients with CIPN had significantly impaired thermal sensitivity at the thenar eminence compared to age matched control participants (Fig 4) with lower CDTs ($ME_{diff}$=4.13°C [0.73, 8.30], U=27.50, $p=0.004$, $r=0.67$) and higher WDTs ($ME_{diff}$=2.63°C [0.47, 5.47], U=131.00, $p=0.018$, $r=0.55$). However, the differences between the two groups were not significant at the dorsum of the foot both for CDT ($ME_{diff}$=3.87°C [−1.50, 16.70], U=66.50, $p=0.369$, $r=0.21$) and WDT ($ME_{diff}$=1.10°C [−2.43, 5.07], U=97.50, $p=0.521$, $r=0.15$).

The results of these experiments provide a rationale for testing thermal sensitivity at the thenar eminence.

## 3.2. Haptic actuator testing and validation

**3.2.1. Age and body site differences in vibration sensitivity.** *When we discussed the findings from the thermal testing experiments, our PPI group reminded us of the potential advantage of conducting all testing at a single site to simplify the process for patients and speed the conduct of testing. This motivated us to compare the vibration sensitivity of the foot dorsum and the thenar eminence.*

**Table 2. Demographics of people with CIPN and control participants.**

|  | CIPN patients (n=13) | Controls (n=13) |
|---|---|---|
| **Sex** |  |  |
| Male | 5 | 7 |
| Female | 8 | 6 |
| **Age** |  |  |
| Mean | 55.69 | 55.00 |
| Range | 28–75 | 28–69 |
| **Cancer** |  |  |
| Breast | 7 |  |
| Leukaemia | 2 |  |
| Lymphoma | 1 |  |
| Oesophagus | 1 |  |
| Pancreatic | 1 |  |
| Testicular | 1 |  |
| **Chemotherapy agent** |  |  |
| Platin | 2 |  |
| Taxane | 5 |  |
| Platin+Taxane | 3 |  |
| Other | 3 |  |

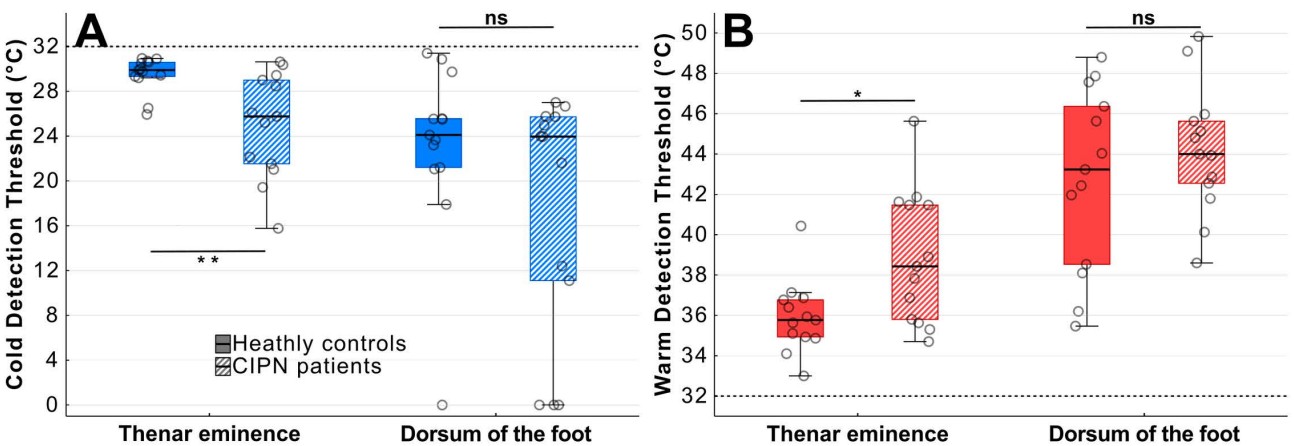

**Fig 4. Thermal thresholds of people with CIPN and healthy age matched controls at the thenar eminence (A) and dorsum of the foot (B).** Healthy controls were more sensitive to temperature changes both for cold detection (CDT) and warm detection (WDT) at the thenar but not the dorsum (*p<0.05; **p<0.01).

Vibration detection thresholds were measured using a haptic actuator and the calibrated tuning fork at the dorsum of the foot and thenar eminence in a group of younger (N=25, 13 female; $M_{age}$=20.84, Min-Max=20–22) and older (N=18, 11 female; $M_{age}$=58.39, Min-Max=50–69) participants.

Analysis of VDTs measured with the haptic actuator revealed two key findings (repeated measures ANOVA – 2 (age-group) by 2 (body site)). Firstly, there was a large main effect of body site ($F_{(1, 41)}$ = 129.20, $p<0.001$, $\eta_p^2$=0.76). The feet were significantly less sensitive to vibration irrespective of age group (Fig 5). And secondly, there was an age by body site interaction ($F_{(1, 41)}$ = 13.88, $p<0.001$, $\eta_p^2$=0.25). Post-hoc comparisons showed that older participants were less sensitive to vibration than younger participants at the dorsum of the foot ($t_{(41)}$ = 3.94, $p<0.001$, d=1.22 [0.55, 1.87]), but no*t* at the thenar eminence ($t_{(41)}$ = 1.25, $p$=1, d=0.39 [−0.23, 0.99]).

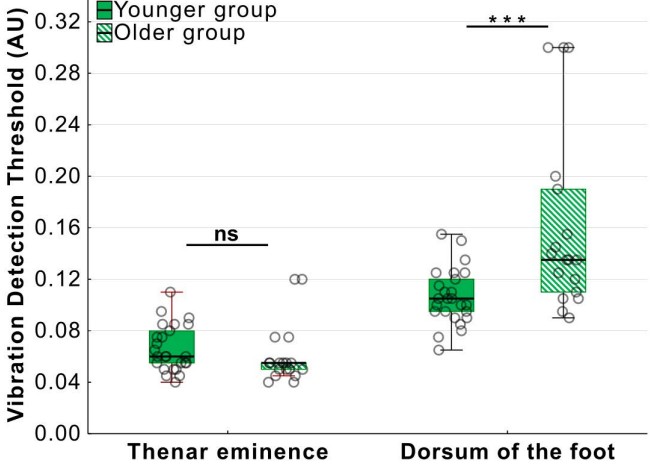

**Fig 5. Vibration thresholds measured using haptic actuator by age-group and test site.** Younger participants had lower vibration detection thresholds (VDT) compared to older participants at the dorsum of the foot but not at the thenar eminence (***p<0.001).

**Table 3. Contingency table for vibration detection measured with the tuning fork.**

| | Score = 8 | Score<8 |
|---|---|---|
| **Thenar eminence** | | |
| Younger group | 19 | 6 |
| Older group | 18 | 0 |
| **Dorsum of the foot** | | |
| Younger group | 20 | 5 |
| Older group | 16 | 2 |

In contrast, the tuning fork shows a lack of test sensitivity because of a floor effect, most of the participants score an average of 8 (top score indicating maximum sensitivity to vibration) at both the thenar eminence and foot locations in all trials; 86% of all participants score an average of 8 at the thenar eminence and 84% do so at the dorsum of the foot. As such, there is virtually no data distribution, therefore, we recoded the data as categorical depending on whether they scored an 8 (Table 3).

There were no significant differences between younger and older participants at the thenar eminence ($\chi^2(1, N=43)$ = 3.22, $p=0.073$) or the dorsum of the foot ($\chi^2(1, N=43)$ = 0.13, $p=0.719$) (Yates corrected Chi-squared tests). Further, there was no significant difference in VDT between the two body areas ($\chi^2(1, N=43)$ = 0.04, $p=0.850$; Yates-corrected McNamar test). Therefore, the testing with the tuning fork did not show the same age and site related differences noted with the haptic stimulation.

Additionally, regardless of other factors in the experimental design, we noted that vibration thresholds were lower when measured using ascending amplitude ramps (M=0.09 AU [0.08, 0.10]) compared to descending ramps (M=0.10 AU [0.09, 0.11]) with the haptic actuator ($F(1, 41)$ = 14.31, $p=0.001$, $\eta_p^2=0.26$).

**3.2.2. Detection of sensory loss with the haptic actuator.** We were interested to know whether the different methods to test vibration sensitivity could track sensory loss. Twelve participants (7 female; $M_{age}=27.83$, Min-Max=20–57) completed a repeated-measures study in which vibration thresholds were measured at the dorsum of the foot using the calibrated tuning fork and the haptic actuator before and after lidocaine block of the superficial peroneal nerve. To track the onset of the analgesic block, we assessed pain ratings in response to application of a 512mN pinprick stimulator (Fig 6). This showed a significant effect of local anaesthetic block on pain ratings ($F(2.18, 23.95)$ = 26.59, $p<0.001$, $\eta_p^2=0.71$, one-way RM-ANOVA with Greenhouse-Geisser correction). Post-hoc comparisons showed a significant reduction in pain ratings after lidocaine injection between 20 and 60 minutes ($p<0.001$) and with a partial recovery from analgesic block at the 60 minute time point ($p=0.004$).

Analysis of change in VDTs measured with the haptic actuator after local anaesthetic block revealed a significant effect of time of measurement ($F(2.17, 23.84)$ = 31.49, $p<0.001$, $\eta_p^2=0.74$) (one-way RM ANOVA with Greenhouse-Geisser correction). Post-hoc comparisons showed that VDTs increased significantly at all post-lidocaine measurements when compared to baseline (20 minutes – d=1.85 [0.88, 2.78]; 40 minutes – d=2.34 [1.21, 3.45]; 60 minutes – d=2.24 [1.15, 3.32]; all $t(11) > 6.16$, all $p<0.001$; Fig 6).

Further, the detection thresholds were lower when measured in response to ascending rather than descending ramps. The nerve block produced a stronger effect on ascending than descending ramps although it was significant for both (Ascending – $F(2.25, 24.74)$ = 54.55, $p<0.001$, $\eta_p^2=0.83$; Descending – $F(2.23, 34.58)$ = 9.91, $p<0.001$, $\eta_p^2=0.47$).

Calibrated tuning fork measures did not provide adequate distributions for conventional analysis (similar to **3.2.1**). In fact, 10 out of the 12 participants achieved a maximum score of 8 when measuring with the tuning fork in every trial at every stage of the experiment. Friedman's non-parametric ANOVA revealed that the observed small changes after blockade were not significant ($\chi^2(3, N=12)$ = 6.69, $p=0.082$).

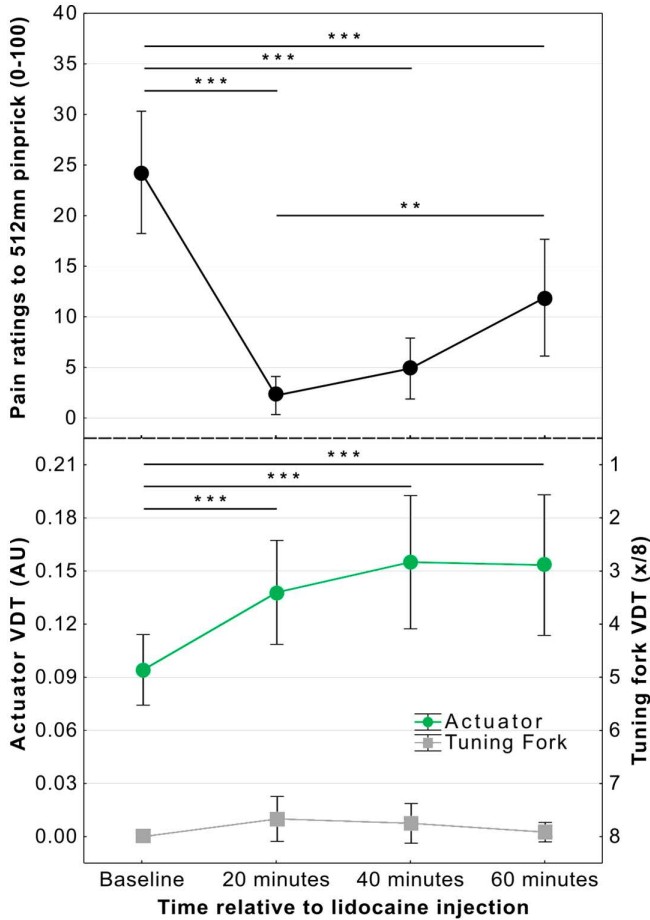

**Fig 6. Effect of superficial peroneal nerve blockade on pain ratings (top), and vibration detection thresholds measured with the haptic actuator and tuning fork (bottom).** Pain ratings were significantly reduced at every time point after lidocaine injection (top). Vibration detection, as measured by the haptic actuator (averaged across ascending and descending ramps) showed a significant loss of sensory function at each time point after blockade. Tuning fork measurements did not identify any change in sensory function (values inverted on scale as higher scores reflect greater sensitivity to vibration at lower amplitudes). Graph shows means±95% CI; **p<0.01; ***p<0.001.

## 4. Discussion

The overarching objective of the SenseCheQ patient partner informed research project is to identify a reliable quantitative sensory testing approach that people having cancer treatment could feasibly administer to themselves to detect chemotherapy-induced damage to nerves. Our patient partners guided and challenged us through this process with their insights from their lived experience of cancer treatment and its long term sequalae. In this series of studies, we have aimed to meet both their requirements and those defined by the pathological processes of CIPN. We sought to identify the most promising sensory test modalities and parameters, the optimal body area(s) for testing, and to establish the validity of a deploying a haptic actuator to assess vibration detection.

Our patient partners indicated that a protocol lasting more than 10–15 minutes would represent an excessive burden on people undergoing chemotherapy which would impact compliance. This led us to seek to define a limited core battery of tests. The current evidence (albeit limited) suggests that that tests of thermal and mechanical detection thresholds are the most sensitive indices of chemotherapy-induced nerve damage [34]. Therefore, we assessed warm, cold and vibration

detection modalities which reflect C-, Aδ- and Aβ fibre function, respectively. This limited set of tests as a battery was anticipated to be relatively rapid to conduct, to be well tolerated and to span the likely range of fibres affected by CIPN. Patient partners emphasized that testing should ideally not require fastening a peripheral probe to the foot and suggested the sole of the foot as a possible solution. Accordingly, we assessed thermal and mechanical sensitivity at the sole and dorsum of the foot. We found that the sole of the foot was too insensitive to changes in temperature and that the variance was much higher than at the dorsum of the foot (in agreement with previous studies of thermo-sensation [42,43]). Even though there was preserved mechanical detection the relative thermal insensitivity even in healthy young participants undermines the viability of the sole as a test site.

Considering these results on the foot, we explored testing the hands which our patient partners encouraged for reasons of convenience and personal experience of symptoms affecting the hands. However, the symptoms of CIPN typically present earlier, more severely, and persist longer at the feet [44–47]. This leads most CIPN studies to focus on sensory function in the feet. Given this trade off, we compared thermal sensitivity at the dorsum of the foot and thenar eminence of the hand using a relatively small thermode size (pragmatically chosen to allow reliable testing on the thenar eminence even in people with smaller hands). We found significantly greater sensitivity, lower variance and minimal age-related deterioration at the thenar eminence compared to the foot. This pattern is in line with previous findings suggesting that the dorsum of the foot is less sensitive to thermal stimulation and more susceptible to age-related sensory loss in heathy participants [42,43,48–50]. We extended the investigation to include a population of people with CIPN and found that we could detect differences in sensory function between people with CIPN and age matched control participants at the thenar, but not at the foot dorsum, indicating a greater power to identify loss of sensory function. We note that multiple previous studies have also been able to identify chemotherapy-induced sensory loss in the hand [26,29,51–55] although only one research group has previously used the thenar eminence as a test site [29,52–54]. As an aside, when assessing warm detection at the dorsum of the foot, many healthy (older) participants as well as patients with CIPN reported hot pain as the only sensation. As previously shown, warm detection thresholds increase, while heat pain thresholds decrease with age [56,57] leading to detection and pain thresholds getting closer and indeed may be indistinguishable for some people in the extremities of the lower limb.

To test large fibre function we aimed to assess the use of an encapsulated linear resonant haptic actuator to deliver vibration stimuli in comparison with the clinical standard calibrated tuning fork. We reasoned that a valid vibration detection measurement protocol should be able to detect differences in vibration stimulation sensitivity between body sites in healthy individuals as well as identify the known age-related decrease of sensitivity [58–61]. Further, we modelled neuropathy by inducing a nerve block and then tested whether each method could detect changes in sensitivity over time. Both experiments showed that the calibrated tuning fork is not sensitive enough to detect these changes due to a floor effect; whereby a large majority of participants recording the maximum sensitivity to vibration on the eight-point scale. In contrast, use of the haptic actuator demonstrated a greater sensitivity to vibration stimulation at the thenar versus foot dorsum, the presence of age-related changes at the foot but not the hand, and the loss of large-fibre mediated sensory function due to nerve blockade.

These vibration testing results further support the rationale for testing at the thenar eminence because of its higher sensitivity, lower variance and less age-related deterioration in function than at the foot. An additional finding, made possible by the ability of the haptic actuator to deliver ascending and descending vibration ramping stimuli (unlike the tuning fork), was that ascending ramps produced lower thresholds with participants displaying greater sensitivity. Consequently, this ascending threshold better tracked changes is sensitivity due to location and simulated sensory loss. This is likely due to a perceptual accommodation effect during the descending ramps. The phenomenon has been documented in other areas of human perception; for example, research into flicker fusion frequency [62]. Additionally, the time taken for participants to detect an ascending ramp was usually much shorter than descending ramps (especially in healthy participants – typically 1-3 s versus 20–25 s per ramp) given that the latter have to start at a level that would be regarded as overtly pathological if undetected at maximum amplitude.

The experiments presented here have some limitations as they are a series of iterative, proof of concept studies and all the experiments were conducted on relatively small numbers of participants. However, the sensitivity analyses and the magnitude of the effects we observed show the studies reached adequate statistical power. We intentionally employed a broad set of inclusion criteria and studied a relatively heterogenous population of people with CIPN associated with different systemic anti-cancer treatments and primary cancer sites across a wide range of ages. Our final choice of sensory modalities is narrow and excludes tests of nociceptive function. It does not comprehensively assess all of the known primary afferent fibre types, nor does it assess sensory function relating to alterations in central processing. This narrowed focus was necessary in order to meet our patient partner informed brief of a compact test protocol to track the known neuropathology of CIPN.

The combination of findings from these studies provide support for testing at the hand, with a smaller thermal probe than typically used and recommended in previous research [27,45], and with a haptic actuator to deliver ascending vibration ramps. Although the choice of test site at the thenar eminence is somewhat uncommon in the CIPN literature – it represents a "sweet spot" in the body's perceptual function for these modalities. It has a dense sensory innervation that enables it to be sensitive to both warm, cold and mechanical stimuli. The gradient of thermal sensitivity declines towards the fingertips [43,50] and sensitivity to vibration increases more distally down the limbs due to increased density of innervation [63–65]. It is a territory with a good blood supply ensuring its temperature is maintained closer to core body temperature than the distal digits. With the added convenience of using an easy to access and manipulate test site, of considerable importance to our patient population, it therefore represents a good candidate region to test for loss of function in neuropathies.

To achieve our overarching aim of enabling home-based self-testing of sensory function longitudinally during courses of systemic anti-cancer therapy lasting many months, we will need to develop equipment that allows these test modalities to be assayed in a simple, convenient, reliable and robust protocol. Given the parameter test framework established within the current studies it will need to be capable of detecting sub-micron level changes in vibration sensitivity and assess thermal thresholds to tenths of a degree while operating in a range of different home environments. This, as a combined package, has not yet been achieved in any medical or research device to date but findings presented here serve as important insights as we continue development towards that goal.

## Acknowledgments

Special thanks to our patient partner group who have contributed to each step of SenseCheQ development and testing programme, (Alan Young, Iain MacLeod, Rubina Zafar, and the other members of the group who have elected to remain anonymous).

Generative AI was not used at any point in conduct or write up of this manuscript.

## Author contributions

**Conceptualization:** Marin Dujmović, James Philip Dunham, Johannes Gausden, Bethany Groves, Elizabeth Anne Cottuli de Cothi, Chris Geddie, Lottie Adams, Alex Grant, Nisla Maharjan, Bhushan Thakkar, Anthony O'Neill, Alan Young, Roger Graham Whittaker, Lesley Colvin, Anthony Edward Pickering.

**Data curation:** Marin Dujmović, Bethany Groves, Elizabeth Anne Cottuli de Cothi, Charlotte Burgess, Chris Geddie, Lottie Adams, Alex Grant, Nisla Maharjan.

**Formal analysis:** Marin Dujmović.

**Funding acquisition:** Anthony Edward Pickering.

**Investigation:** Marin Dujmović, Bethany Groves, Elizabeth Anne Cottuli de Cothi, Charlotte Burgess, Chris Geddie, Lottie Adams, Alex Grant, Nisla Maharjan.

**Methodology:** Marin Dujmović, James Philip Dunham, Bethany Groves, Elizabeth Anne Cottuli de Cothi, Chris Geddie, Lottie Adams, Alex Grant, Nisla Maharjan, Alan Young, Anthony Edward Pickering.

**Resources:** Johannes Gausden.

**Software:** Marin Dujmović, Johannes Gausden.

**Supervision:** Marin Dujmović, James Philip Dunham, Anthony O'Neill, Roger Graham Whittaker, Lesley Colvin, Anthony Edward Pickering.

**Visualization:** Marin Dujmović.

**Writing – original draft:** Marin Dujmović, Anthony Edward Pickering.

**Writing – review & editing:** Marin Dujmović, James Philip Dunham, Johannes Gausden, Bethany Groves, Elizabeth Anne Cottuli de Cothi, Charlotte Burgess, Chris Geddie, Lottie Adams, Alex Grant, Nisla Maharjan, Bhushan Thakkar, Anthony O'Neill, Alan Young, Roger Graham Whittaker, Lesley Colvin, Anthony Edward Pickering.

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
