## [Decision Letter · Decision Letter 0]

30 Oct 2025

Dear Dr. Dujmović,

Thank you for submitting your manuscript to PLOS ONE. After careful consideration, we feel that it has merit but does not fully meet PLOS ONE’s publication criteria as it currently stands. Therefore, we invite you to submit a revised version of the manuscript that addresses the points raised during the review process.

We look forward to receiving your revised manuscript.

Kind regards,

Armando Almeida

Academic Editor

PLOS ONE

Journal Requirements:

The SenseCheQ research project was funded as part of the Advanced Pain Discovery Platform by a consortium made up of UKRI MRC, Versus Arthritis, and Eli Lilly all coordinated by the MRC [MR/W027925/1].

BT was also supported by an APDP grant as part of the Partnership for Assessment and Investigation of Neuropathic Pain: Studies Tracking Outcomes, Risks and Mechanisms (PAINSTORM) consortium [MR/W002388/1].

4. Please expand the acronym “MRC” (as indicated in your financial disclosure) so that it states the name of your funders in full.

Special thanks to our patient partner group who have contributed to each step of SenseCheQ development and testing programme, (Alan Young, Iain MacLeod, Rubina Zafar, and the other members of the group who have elected to remain anonymous). The SenseCheQ research project was funded as part of the Advanced Pain Discovery Platform by a consortium made up of UKRI MRC, Versus Arthritis, and Eli Lilly all coordinated by the MRC [MR/W027925/1].

BT was also supported by an APDP grant as part of the Partnership for Assessment and Investigation of Neuropathic Pain: Studies Tracking Outcomes, Risks and Mechanisms (PAINSTORM) consortium [MR/W002388/1].

Anonymized data and R markdown code for reproducing analyses is available on the Open Science Framework at http://doi.org/10.17605/OSF.IO/F5T82. AEP declares consultancy work for Lateral Pharma and has a research grant from Eli Lilly on an unrelated topic. All other authors have no conflicts of interest to declare.

Generative AI was not used at any point in conduct or write up of this manuscript.

The SenseCheQ research project was funded as part of the Advanced Pain Discovery Platform by a consortium made up of UKRI MRC, Versus Arthritis, and Eli Lilly all coordinated by the MRC [MR/W027925/1].

BT was also supported by an APDP grant as part of the Partnership for Assessment and Investigation of Neuropathic Pain: Studies Tracking Outcomes, Risks and Mechanisms (PAINSTORM) consortium [MR/W002388/1].

6. Thank you for stating the following in the Competing Interests/Financial Disclosure section:

AEP declares consultancy work for Lateral Pharma and has a research grant from Eli Lilly on an unrelated topic. All other authors have no conflicts of interest to declare.

We note that one or more of the authors are employed by a commercial company: Lateral Pharma and Eli Lilly

Reviewers' comments:

Reviewer's Responses to Questions

**Comments to the Author**

1. Is the manuscript technically sound, and do the data support the conclusions?

Reviewer #1: Yes

Reviewer #2: Yes

2. Has the statistical analysis been performed appropriately and rigorously?

Reviewer #1: Yes

Reviewer #2: Yes

3. Have the authors made all data underlying the findings in their manuscript fully available?

Reviewer #1: Yes

Reviewer #2: Yes

4. Is the manuscript presented in an intelligible fashion and written in standard English?

Reviewer #1: Yes

Reviewer #2: Yes

Reviewer #1: The authors present novel data related to the initial development of a sensory testing protocol, with the aim of providing a simple test kit that can be used for longitudinal self-administered assessments by patients at home. This has significant clinical relevance for the early recognition and monitoring of chemotherapy-induced neuropathy. The manuscript is well-written and the Figures are clear. Data related to differences based on age and body site have broader relevance to QST methodology and results.

A particular strength of the study is the detailed input and feedback from patient partners who have informed all stages of development. This is crucial for ensuring at home testing is feasible and acceptable.

Some minor points that could be clarified or included in the Discussion.

Section 2.2: “Experimental sessions were conducted in a controlled clinical study room environment (at the NIHR Bristol Clinical Research Facility) with ambient temperature of 21°C.” Do the authors anticipate any confounding effects related to environmental and/or skin temperature during at home testing? Will further instructions related to ambient temperature be needed for home sensory testing?

Section 2.3.1 Methods for mechanical detection and pain thresholds are not described in the same detail as the thermal and vibration testing. Some additional data could be added (eg. number of trials, range of intensities) for readers not familiar with the DFNS protocol.

Overall, analyses are appropriate for the current stage of development of the test protocol. However, it would have been interesting to evaluate test-retest reliability in volunteers.

2.4.2 The haptic actuator detected sensory loss changes with LA block, but does not appear to have been used in the CIPN group. Recruitment for the vibration study seems to have been earlier. The authors acknowledge that further device testing and development is needed, but do they have evidence that the vibration methodology reported here will also have sensitivity for sensory loss associated with CIPN and be feasible for use at home?

From the final paragraph “Given the parameter test framework established within the current studies it will need to be capable of detecting sub-micron level changes in vibration sensitivity and assess thermal thresholds to tenths of a degree while operating in a range of different home environments. This, as a combined package, has not yet been achieved.”

Is the Conclusion that it will not be feasible to develop equipment which is cost-effective, portable and has adequate sensitivity, or rather that further development work is planned?

Reviewer #2: This is a clear manuscript reporting valuable results.

My only point relates to the figures - the authors mostly use bar graphs for data which is not percentages. Please use violin-plots, box-dot plots (as you already do in Figure 4) for figures 2,3, and 5.

**Do you want your identity to be public for this peer review?** For information about this choice, including consent withdrawal, please see our Privacy Policy

Reviewer #1: No

Reviewer #2: No

---

## [Author Response · Author response to Decision Letter 1]

10 Nov 2025

Reviewer 1

Comment:

The authors present novel data related to the initial development of a sensory testing protocol, with the aim of providing a simple test kit that can be used for longitudinal self-administered assessments by patients at home. This has significant clinical relevance for the early recognition and monitoring of chemotherapy-induced neuropathy. The manuscript is well-written and the Figures are clear. Data related to differences based on age and body site have broader relevance to QST methodology and results.

A particular strength of the study is the detailed input and feedback from patient partners who have informed all stages of development. This is crucial for ensuring at home testing is feasible and acceptable.

Some minor points that could be clarified or included in the Discussion.

Response:

We thank the reviewer for their feedback and encouraging comments.

Comment:

Section 2.2: “Experimental sessions were conducted in a controlled clinical study room environment (at the NIHR Bristol Clinical Research Facility) with ambient temperature of 21°C.” Do the authors anticipate any confounding effects related to environmental and/or skin temperature during at home testing? Will further instructions related to ambient temperature be needed for home sensory testing?

Response:

This is exactly the challenge of moving Quantitative Sensory Testing from the lab to the home environment. As we note in the conclusion, the protocol that solves this challenge will need to operate in a range of different environments. In our upcoming work we will explicitly report what hardware, software and interface solutions were implemented in order to reduce these sources of variance. This work is underway, and a new publication will be upcoming in the near future.

Comment:

Section 2.3.1 Methods for mechanical detection and pain thresholds are not described in the same detail as the thermal and vibration testing. Some additional data could be added (eg. number of trials, range of intensities) for readers not familiar with the DFNS protocol.

Response:

We have added the following text to the section

Mechanical detection was conducted by determining the first filament weight which did not elicit a touch percept, then proceeding in a staircase manner. The procedure ended when five sub- and supra-thresholds values were measured. Mechanical pain was measured in a similar manner; the first pinprick weight that elicited a sharp percept was determined followed by a staircase approach to establish five sub- and supra-threshold values.

Comment:

Overall, analyses are appropriate for the current stage of development of the test protocol. However, it would have been interesting to evaluate test-retest reliability in volunteers.

Response:

Our upcoming work will feature test-retest of our home sensory testing protocol with healthy adults as well as performance testing and description of hardware, software, and interface solutions.

Comment:

2.4.2 The haptic actuator detected sensory loss changes with LA block, but does not appear to have been used in the CIPN group. Recruitment for the vibration study seems to have been earlier. The authors acknowledge that further device testing and development is needed, but do they have evidence that the vibration methodology reported here will also have sensitivity for sensory loss associated with CIPN and be feasible for use at home?

Response:

These are all important questions which we tackle in our upcoming work on how our sensory testing device performs. The current paper shows that 1) the clinical standard tuning fork lacks sensitivity, 2) that vibration sensitivity is largely unaffected by age-related change at the thenar (but not foot) and 3) that using a linear resonant actuator allows demonstration of significant differences (age-related at the foot and nerve block) – which makes it a more promising solution.

Comment:

From the final paragraph “Given the parameter test framework established within the current studies it will need to be capable of detecting sub-micron level changes in vibration sensitivity and assess thermal thresholds to tenths of a degree while operating in a range of different home environments. This, as a combined package, has not yet been achieved.”

Is the Conclusion that it will not be feasible to develop equipment which is cost-effective, portable and has adequate sensitivity, or rather that further development work is planned?

Response:

We have added text to the end of the sentence to make it clear development is underway and that findings presented in the study provide valuable insights towards achieving this goal. The expanded sentence states:

This, as a combined package, has not yet been achieved in any medical or research device to date but findings presented here serve as important insights as we continue development towards that goal.

Reviewer 2

Comment:

This is a clear manuscript reporting valuable results.

My only point relates to the figures - the authors mostly use bar graphs for data which is not percentages. Please use violin-plots, box-dot plots (as you already do in Figure 4) for figures 2,3, and 5.

Response:

We thank the reviewer and have now changed Figures 2, 3 and 5 to match the box-plot + data points style from Figure 4.

---

## [Editor Report · Decision Letter 1]

18 Nov 2025

Sense-checking the approach to quantitative sensory testing to detect chemotherapy-induced peripheral neuropathy

PONE-D-25-49880R1

Dear Dr. Dujmović,

We’re pleased to inform you that your manuscript has been judged scientifically suitable for publication and will be formally accepted for publication once it meets all outstanding technical requirements.

Kind regards,

Armando Almeida

Academic Editor

PLOS ONE
---

## [Editor Report · Acceptance letter]

PONE-D-25-49880R1

PLOS ONE

Dear Dr. Dujmović,

I'm pleased to inform you that your manuscript has been deemed suitable for publication in PLOS ONE. Congratulations! Your manuscript is now being handed over to our production team.

Kind regards,

on behalf of

Prof. Armando Almeida

Academic Editor

PLOS ONE